# Diversity of Culturable Bacteria from the Coral Reef Areas in the South China Sea and Their Agar-Degrading Abilities

**DOI:** 10.3390/microorganisms12010187

**Published:** 2024-01-17

**Authors:** Mei Liu, Fu Yin, Wenbin Zhao, Peng Tian, Yi Zhou, Zhiyu Jia, Keyi Huang, Yunqi Ding, Jiaguang Xiao, Wentao Niu, Xiaolei Wang

**Affiliations:** 1College of Marine Life Sciences, Frontiers Science Center for Deep Ocean Multispheres and Earth System, Ocean University of China, 5 Yushan Road, Qingdao 266003, China; liumeihaiyang@163.com (M.L.); yinfuhaida@163.com (F.Y.); zhaowenbin926@163.com (W.Z.); zhouyi7370@163.com (Y.Z.); huangkeyi226@163.com (K.H.); dingyunqi000826@163.com (Y.D.); 2Institute of Evolution & Marine Biodiversity, Ocean University of China, Qingdao 266100, China; 3Laboratory for Marine Ecology and Environmental Science, Laoshan Laboratory, Qingdao 266071, China; 4Laboratory of Marine Biodiversity Research, Third Institute of Oceanography, Ministry of Natural Resources, 178 Daxue Road, Xiamen 361005, China; tianpeng@tio.org.cn (P.T.); jiazhiyu23@mails.ucas.ac.cn (Z.J.); xiaojiaguang@tio.org.cn (J.X.); 5Nansha Islands Coral Reef Ecosystem National Observation and Research Station, Guangzhou 510300, China

**Keywords:** coral reef, diversity, heterotrophic bacteria, *Vibrio*, agarases

## Abstract

The South China Sea (SCS) is abundant in marine microbial resources with high primary productivity, which is crucial for sustaining the coral reef ecosystem and the carbon cycle. Currently, research on the diversity of culturable bacteria in the SCS is relatively extensive, yet the culturable bacteria in coral reefs has been poorly understood. In this study, we analyzed the bacterial community structure of seawater samples among Daya Bay (Fujian Province), Qionghai (Hainan Province), Xisha Islands, and the southern South China Sea based on culturable methods and detected their abilities for agar degradation. There were 441 bacterial strains, belonging to three phyla, five classes, 43 genera, and 101 species, which were isolated by marine agar 2216E (MA; Becton Dickinson). Strains within *Gammaproteobacteria* were the dominant group, accounting for 89.6% of the total bacterial isolates. To investigate vibrios, which usually correlated with coral health, 348 isolates were obtained from TCBS agar, and all isolates were identified into three phylum, three classes, 14 orders, 25 families, and 48 genera. Strains belonging to the genus *Vibrio* had the greatest number (294 strains), indicating the high selectivity of TCBS agar for vibrios. Furthermore, nineteen strains were identified as potentially novel species according to the low 16S rRNA gene similarity (<98.65%), and 28 strains (15 species) had agar-degrading ability. These results indicate a high diversity of culturable bacteria in the SCS and a huge possibility to find novel and agar-degrading species. Our study provides valuable microbial resources to maintain the stability of coral ecosystems and investigate their roles in the marine carbon cycle.

## 1. Introduction

The coral reef is one of the important ecosystems in the ocean, with high primary productivity that supports around one-third of the diversity of marine life, including endolithic algae, bacteria, archaea, fungi, and viruses [1,2]. The near-coral seawater environments where coral and microbial interactions occur is called the “coral biosphere” [3]. Coral mucus, released by reef-building organisms, usually serves as a suitable growth substrate for heterotrophic bacteria in seawater, harboring an abundant microbial community with a consequent increase in carbon flux [4]. The majority of bacteria that have been found in association with corals are heterotrophic, which usually play a crucial and complicated function in sustaining homeostasis and promoting carbon cycling between corals and symbiotic bacteria [5]. The microbial populations in coral reef ecosystems can be altered as a result of environmental perturbations. Large amounts of highly diverse microbes, specifically associated with corals, play a fundamental role in the health of the corals [6]. However, outbreaks of coral diseases, caused by climate change and parthenogenic microbes associated with their tissue, have been blamed for the dramatic decline in coral diversity [7]. Many coral diseases have been reported that are usually caused by some pathogens or disease-causing bacteria. Most coral pathogens are unknown in terms of their characteristics, and several coral diseases are brought on by complex microbial complexes rather than a single species [8]. Of these, the most devastating coral disease is bacterial bleaching of corals. It has been found that for coral bleaching, changes in bacterial community are a major driver. Additionally, a number of other diseases, such as white band and black belt diseases, have been associated with changes in coral-associated microbial assemblages [9]. On one hand, coral-associated microbial communities can be beneficial to the host. They may have positive or facilitating effects on homeostasis and the carbon cycle. On the other hand, changes in host and environmental conditions make them exhibit negative or antagonistic effects on corals [10]. For example, nutritious seawater provides an opportunity for bacteria to thrive, whereas the prosperity of bacteria may trigger the extinction and depression of reef-building corals [11]. 

Coral mucus accommodates a variety of microorganisms at concentrations that are much higher than the surrounding seawater [12,13], and the majority of these are *Vibrio* strains. Most *Vibrio* strains have been found to cause marine organism diseases when mucus detached from the coral, such as *V. mediterranei*, *V. coralliilyticus*, *V. harveyi*, and *V. alginolyticus*, which are the causative agents for coral bleaching [14,15]. Haldar et al. have studied the coral ecosystem associated with *Vibrio* on Kurusadai island and Tamil Nadu, and they have found that the abundance of *Vibrio* species increased with coral bleaching and decreased after the recovery from the bleaching event [16]. *Acropora solitaryensis* in the Xisha Islands have experienced severe bleaching, and the proportion of *Vibrio* in bleached corals are significantly higher than in that of healthy corals [17]. In addition, *Vibrio* spp. have the ability to utilize a wide range of carbon sources and play a significant role in the transmission and circulation of numerous materials and energies in coral reef areas [18], especially chitin, alginate, and agar. Some *Vibrio* species, like *V. parahaemolyticus* and *V. alginolyticus*, can degrade agar by producing the related enzymes [19]. Thus, corals and the surrounding seawater may be ideal sites to study the distribution and pathogenicity of *Vibrio*, as well as to discover novel species and enzymes. 

Recently, the physiological and ecological characteristics of *Vibrio* have been extensively studied by various methods, including molecular and culturable approaches. Based on *Vibrio*-specific 16S rRNA amplicon sequencing, the abundance and distribution pattern of *Vibrio* have been induced by multiple environmental parameters [20]. In the marginal sea of northern China, the abundance of *Vibrio* spp. was significantly higher in summer than that in winter and was affected by temperature, among other factors [21]. Wang et al. reported that the concentration of organic matter in sediments of Chinese marginal seas is significantly higher than that in the water column, and vibrios show spatial heterogeneity with a joint effect of spatial and environmental factors [22]. All the studies have reported on the dynamics of total *Vibrio* and the influences of physicochemical parameters, especially for species that are difficult to isolate from aquatic environments using traditional methods [23]. However, it is impossible to discern *Vibrio* that are in active, dead, or viable-but-none-culturable (VBNC) states based on a single sequencing approach. To detect the functions of *Vibrios*, it is essential to obtain a pure culture of these strains [24]. Further physiological studies of *Vibrio* and the development of related organisms will require the isolation and cultured strains from a variety of environments. For example, in the study of *Vibrio* spp. in Ishigaki coral reef ecosystems, thiosulfate citrate bile salts sucrose agar (TCBS agar) has been used to isolate *Vibrios*, and they have found high *Vibrio* abundance and pathogenic species in the coral surrounding seawater [25].

The abundance and diversity of bacteria in coral reef areas of the South China Sea (SCS) are higher than in non-coral areas, although they also currently face coral disease and bleaching events [26]. The dominant bacteria of different coral species are similar, which were distributed in *Gammaproteobacteria*, *Alphaproteobacteria*, *Firmicutes* [27], *Bacteroidetes*, and *Actinobacteria*. Coral bleaching may be attributed to the dominant *Vibrio* associated with corals [28]. In this study, seawater samples were collected near the coral reefs in the SCS, including Daya bay (DY), Qionghai (QH), Xisha islands (XS), and the Southern South China Sea (SSCS) during May to October 2020. We used marine agar 2216E agar (MA; Becton Dickinson) to isolate heterotrophic bacteria and TCBS agar to isolate coral pathogenic *Vibrio*. In addition, the bacteria were identified based on 16S rRNA gene sequencing and further detected agar degradability. Our results will provide a unique microbial resource base for understanding the diversity of heterotrophic bacteria and will help in investigating their potential roles in coral ecosystems.

## 2. Materials and Methods

### 2.1. Sample Collection

Seawater samples were collected from four different sampling sites, including Daya Bay, Qionghai, Xisha Islands, and the SSCS aboard the R/V *Yuezhanyuke10* during 31 May to October 2020. The seawater samples were collected using Niskin bottles, and sampling stations among different periods are shown in Figure 1.

### 2.2. Bacterial Isolation

The seawater samples were serially diluted from 10^−1^ to 10^−3^ with 0.85% (*w*/*v*) saline, and triplicate 200 µL of each dilution was spread onto the surface of MA plates (1.00 g yeast extract, 5.00 g peptone, 0.01 g ferric phosphate, 20.00 g agar, 1000 mL seawater, pH 7.6) and TCBS agar plates (HopeBiol, Qingdao, China). The plates were cultured at 28 °C, and colonies were randomly picked and purified by streaking three times on MA plates. These strains were preserved at −80 °C with 15% (*v*/*v*) glycerol.

### 2.3. Genomic DNA Extraction

The genomic DNA of the bacterial isolates were extracted by the “colony boiling method”. The bacterial isolate was sub-cultured on an MA agar plate through the plate marking method, and individual colonies were picked into centrifuge tubes containing 200 μL 1 × TE buffer (1 mol/L Tris–HCl, 0.5 mol/l EDTA, pH 8.0). The centrifuge tubes were heated on water at 100 °C for 10 min and immediately ice-bathed for 30 min (mins). After that, the centrifuge tube was centrifuged at 4 °C, 14,000× *g* for 5 min after the ice bath completed, and the supernatants were used as DNA templates.

### 2.4. PCR Amplification for 16S rRNA Genes and Identification for Isolated Microbes

The identification of the isolates relied on 16S rRNA gene amplification using the primers B8F (5′-AGAFTTTGATCCTGGCTCAG-3′)/1510R (5′-GGTTACCTTGTTACGACTT-3′). The reaction system for PCR was 30 μL, including 15 μL TaKaRa Taq, 13 μL ddH_2_O, 0.25 μL of the primer pairs (20 μmol/L), and 1.5 μL of genomic DNA. The PCR program was set as follows: 94 °C for 5 min, and 30 cycles of 94 °C for 60 s, 55 °C for 60 s, and 72 °C for 90 s, and a final extension of 72 °C for 10 min. In order to determine whether the target band was amplified, the PCR products of 16S rRNA gene sequences were put through agarose gel electrophoresis. The PCR products of 16S rRNA gene were sequenced at Beijing Genomics Institute (BGI, Qingdao, China), and then, the sequences were screened for valid fragments (~650 bp) by Chromas [29]. The 16S rRNA gene sequences of the 789 isolates were listed in Appendix A. Sequence similarities between isolates and their most closely related bacteria were calculated using the EzBiocloud server (https://www.ezbiocloud.net; 15 October 2022), and information about the bacterial strains and their similar strains were counted by Excel. The phylogenetic tree was constructed using MAFFT version 7 (https://mafft.cbrc.jp/alignment/server/; 14 December 2022), Gblocks Server (http://molevol.cmima.csic.es/castresana/Gblocks_server.html; 15 December 2022), and Mega-X (Neighbor-joining algorithm) [30] to construct the phylogenetic tree, and iTOL (https://itol.embl.de/login.cgi; 16 December 2022) to further improve the evolutionary tree.

### 2.5. Genomic Sequencing, Assembly, and Functional Annotation

The genomic DNA of *V. agarilyticus* WXL890 (SR4) and *F. agarivorans* WXL884 (SM6) were isolated utilizing the phenol-chloroform-isoamyl alcohol extraction technique. The quantity and quality of genomic DNA were detected using a NanoDrop 2000 spectrophotometer (Thermo Scientific, Waltham, MA, USA). Whole-genome sequencing was performed by Beijing Genomics Institute using the Illumina HiSeq (270-bp inserts library; Illumina, San Diego, CA, USA). The trimmed sequences from the Illumina HiSeq platform were assembled by unicycler version 0.5.0 [31]. Pilon version 1.16 was used to polish Illumina data. Checkm [32] was used to objectively measure the quality of the genome. The prediction of open reading frames and their functional annotations were performed using the RAST server (https://rast.nmpdr.org/rast.cgi; 15 March 2023). Uniprot/Swissprot (https://www.ncbi.nlm.nih.gov; 6 April 2023) was used to find the identity and coverage of protein sequences.

### 2.6. Detection of the Agar-Degrading Abilities of Microbes

The isolated strains underwent the determination of agar degradation. Cells were prepared by growing at 28 °C to the mid-logarithmic growth phase (optical density at 600 nm [OD600] = 0.6) on the MA plate to find whether the medium had depressions.

### 2.7. Analysis of Agarases Based on Nucleotide and Amino Acid Sequences

To explore the phylogenetic and molecular evolutionary relationships, the enzymes in each family of the agar-degrading genes were randomly selected from carbohydrate-active enzymes (CAZy) database. The amino acid sequences were downloaded from the Genbank database to build an agar-degrading database with those agar-degrading genes in strains SR4 and SM6. Multiple sequences were compared by CLUSTALW software (Multiple Sequence Alignment—CLUSTALW (https://www.genome.jp); 20 April 2023) [33]. After sequencing, the phylogenetic and molecular evolutionary relationships of agarase coding gene were analyzed by MEGA-X. According to the CAZY database (http://www.cazy.org/; 15 April 2023), GH16, GH116, GH50, GH86, and GH118 families were elected to phylogenetic analysis with SR4 (*aga1177*, *aga1178*) and SM6 (*aga1265*, *aga1271*, *aga1293*, *aga1295*, *aga1297*, and *aga1299*).

## 3. Results

### 3.1. Diversity of Heterotrophic Bacteria among All Samples

A total of 441 strains were isolated from the MA medium, including 81 strains from Daya Bay, 98 strains from Qionghai, 104 strains from the Xisha Islands, and 158 strains from the SSCS (Figure 1). According to the similarity analyses of 16S rRNA genes, all the strains belonged to three phyla, five classes, 43 genera, and 101 species. The three phyla were *Proteobacteria* (416 strains, consisting of *Gammaproteobacteria* and *Alphaproteobacteria*), *Bacteroidetes* (23 strains), and *Firmicutes* (2 strains). At the class level, *Gammaproteobacteria* was the dominant group, including 365 strains (belonging 69 species within 19 genera) and accounting for 82.8%, followed by *Alphaproteobacteria* with a high relative abundance (accounting for 11.7%). Consistently, the remaning class included only 13 strains belonging to 6 species in 4 genera of *Flavobacteriia*, ten strains belonged to 4 species in 3 genera of *Cytophagia*, and two strains of *Bacilli* (Figure 2A).

At the genus level, 172 strains were isolated from 27 different species; *Vibrio* (*Gammaproteobacteria*) had the greatest number of strains and made up 39% of all isolates. *Pseudoalteromonas* (*Gammaproteobacteria*, accounting for 20.9%) was the second-largest genus with 92 strains affiliated with 11 species. *Alteromonas*, the third-largest genus with 41 strains of 6 species, was not isolated from Qionghai. Furthermore, the numbers of *Tritonibacter*, *Idiomarina*, *Halomonas*, and *Tenacibaculum* strains were also relatively high, followed by 21 strains of 5 species, 16 strains of 4 species, 13 strains of 4 species, and 11 strains of 3 species, respectively. Some other genera, such as *Marinobacter*, *Flammeovirga*, *Pseudovibrio*, and *Photobacterium* contained 4~10 strains. There were also a lot of genera that only contained a single isolate, reflecting the diversity of the microbiome in the SCS. Among them, *V. neocaledonicus* was the dominant species with 49 strains, and *Alteromonas macleodii*, *V. owensii*, and *Pseudoalteromonas shioyasakiensis* had high relative abundance as well (Figure 2B).

In order to know the community structure of coral pathogens (i.e., *Vibrio*), TCBS medium was chosen to do further analyses. There were 348 strains isolated from the TCBS agar, including 50 species within 14 genera (Figure 3). At the class level, it was shown that the culturable bacteria mainly affiliated with *Gammaproteobacteria* (342 strains), *Alphaproteobacteria* (5 strains), and *Bacilli* (1 strain). Above all, *Vibrio* was the absolutely dominant genus with 294 strains belonging to 28 species, accounting for 84.5%. Similar to the MA results, *V. neocaledonicus* was the most abundant culturable bacteria with 88 strains, followed by *V. owensii* with 45 strains and *V. campbellii* with 30 strains. Furthermore, there were also quite high numbers of *V. maritimus* (18 strains), *V. harveyi* (16 strains), *V. azureus* (15 strains), and *V. parahaemolyticus* (13 strains). Regarding the other genera, the abundance of *Photobacterium*, *Pseudoalteromona*, and *Halomonas* was relatively high, including 13 strains of 4 species, 13 strains of 3 species, and 11 strains of 3 species, respectively (Figure 3B). On the contrary, the diversity and abundance isolated on TCBS medium were significantly lower than those from MA.

### 3.2. Differences of Heterotrophic Bacteria among Different Locations

The diversity of culturable microorganisms varied significantly across different geographical areas. Among all 441 strains of bacteria isolated from the MA medium, 81 strains belonged to 20 species, and 8 genera were isolated from Daya Bay, 98 strains belonged to 44 species, and 23 genera were isolated from Qionghai, 104 strains belonged to 39 species, and 19 genera were isolated from Xisha Islands, and 158 strains belonged to 46 species, and 17 genera were isolated from SSCS. The SSCS was the most diverse and abundant of bacteria isolated location, whereas the lowest one was Daya Bay. Additionally, *Bacteroidetes* (20 strains) had the highest abundance in Qionghai, whereas the other three regions contained only one or two strains. At the genera level, *Vibrio*, as a dominant genus with high distribution in Daya Bay (32 strains), Qionghai (40 strains), Xisha Islands (22 strains), and the SSCS (78 strains). Consistently, *Pseudoalteromonas* was the second dominant genera with higher abundance among all regions, especially In the SSCS. *Alteromonas* and *Idiomarina* had the highest abundance in the Xisha Islands, whereas *Tenacibaculum* and *Flammeovirga* were only isolated from Qionghai. These results reflected the diversity and abundance of microorganisms in various sea areas. Some dominant bacterial species, such as *V. neocaledonicus*, *Alteromonas macleodii*, *Pseudoalteromonas piscicida*, and *Pseudoalteromonas ruthenica*, were discovered in all the four sampling sites (Figure 4). 

Of the 294 strains of *Vibrio* isolated from TCBS medium, 91 strains belonged to 15 species were isolated from Daya Bay, 23 strains belonged to 9 species were isolated from Qionghai, 103 strains belonged to 24 species were isolated from Xisha Islands, and 77 strains belonged to 17 species were isolated from the SSCS. *V. neocaledonicus,* was the dominant species and had high abundance in Daya Bay and the SSCS. *V. owensii* was the second most abundant species and had high abundance in Xisha Islands and the SSCS. In addition, *V. campbellii*, *V. parahaemolyticus*, and *V. harveyi* showed relatively high abundance in Daya Bay, Xisha Islands, and the SSCS, respectively. *V. parahaemolyticus* was the dominant group in Xisha Islands, whereas *V. owensii* and *V. harveyi* had high relative abundance in the SSCS and Xisha Islands. Combining *Vibrio* (466 strains) isolates from MA and TCBS agars, Xisha Islands had the most diverse isolates with 27 species and the SSCS had the highest abundance of *Vibrios* with 155 strains, whereas Qionghai showed the rarest diversity and abundance with 63 strains belonging to 15 species.

### 3.3. Potentially Novel Bacterial Species Isolated from the Coral Areas

Nineteen bacterial strains (2.4% of the total isolates) showed low 16S rRNA gene similarities (less than 98.65%) to the type strains of their closest known bacterial species, representing 19 potential novel species (Figure 5). Two of these have been recently published by our laboratory, namely *Flammeovirga agarivorans* SR4^T^ and *V. agarilyticu* SM6. All the strains were isolated from different media, i.e., 14 strains (3.2% of the total isolates) were isolated on MA medium, including Qionghai (12 strains) and the SSCS (2 strains). Additionally, 5 strains were isolated from TCBS medium, including Daya Bay (1 strain), Qionghai (1 strain), Xisha Islands (2 strains) and the SSCS (1 strain). Thus, a total of 19 potential novel species belonged to two phyla, four classes, and 10 genera. The two phyla were *Proteobacteria* (16 strains, consisting of *Gammaproteobacteria* and *Alphaproteobacteria*) and *Bacteroidetes.* Among them, the highest proportion of potential novel species belonged to *Gammaproteobacteria* (12 strains), accounting for 63.2%; the remaining strains were mainly affiliated with *Alphaproteobacteria* (4 strains), *Flavobacteria* (2 strains), and *Cytophagia* (1 strain). At the genus level, *Vibrio* had the highest number of potential novel species, accounting for 47.4%, and were classified into the Harveyi, Agarivorans, Mediterranei, Coralliilyticus, Aestivus, Marisflavi clades. Other genera included *Thalassotalea*, *Tencibaculum*, *Flammeovirga*, and *Flavivirga*, etc. (Figure 5).

### 3.4. The Abilities of Bacterial Isolates to Agar-Degrading and the Phylogenetic of Agarases

The agar degradation abilities of total strains isolated from the SCS were investigated. Twenty-two strains had the ability of agar degradation among all 411 strains, of which 6 strains from Daya Bay, 8 strains from Qionghai, and 8 strains from the SSCS. Furthermore, six strains with agar-degrading activity were isolated from TCBS medium, including 1 strain from Qionghai, 3 strains from Xisha Islands, and 2 strains from the SSCS. Qionghai had the highest agar-degrading strains, accounting for 35.7%. These strains belong to *Gammaproteobacteria* (*Proteobacteria*, 22 strains) and *Cytophagia* (*Bacteroidetes*, 6 strain), including *Vibrio* (13 strains), *Pseudoalteromonas* (6 strains), *Flammeovirga* (6 strains), *Marinobacter* (1 strain), *Alteromonas* (1 strain), and *Agarivorans* (1 strain), respectively. Interestingly, all the agar-hydrolyzing strains of *Flammeovirga* were isolated from Qionghai (Figure 6).

Among the isolates, strains SR4 and SM6 showed stronger agar-degrading activities. Genomic analyses showed that strain SR4^T^ maintained two agar lyase-encoding genes (*aga1177*, *aga1178*), and strain SM6 had six potential agar lyase-encoding genes (*aga1265*, *aga1271*, *aga1293*, *aga1295*, *aga1297*, *aga1299*). Members of the glycoside hydrolase (GH) families 50, 116, 118, 16, and 86 were selected for gene construction in strains SR4 and SM6. For strain SR4^T^, *aga1177* belonged to GH16 family, and *aga1178* belonged to GH86 family. In strain SM6, *aga1297*, *aga1293*, *aga1271*, *aga1295*, and *aga1299* belonged to GH50, whereas *aga1265* belonged to GH116 (Figure 7).

## 4. Discussion

The coral reef ecosystem provides abundant and various microbial communities that affect ecosystem processes and host physiology that play a major role in biogeochemical cycles [34]. Although culture-independent methods (16S rRNA gene high-throughput sequencing and metagenomics) have been widely used to study the diversity and potential metabolic pathways of environmental microorganisms, pure culture studies are still necessary for a deep understanding of life characteristics and functions [35]. In this study, a large number of heterotrophic bacteria were isolated from the SCS through large-scale isolation and identification, which reflects the complex community composition of culturable microbes in coral reef areas and their potential ability to degrade agar.

### 4.1. Highly Diverse Heterotrophic Bacteria Survived in the Coral Reef Areas

The coral reef ecosystem maintains high primary productivity and can provide various nutrients for microbes [36]. In this study, a total of 441 strains (belonging to 43 genera and including 50 potential novel species) were isolated from four regions (Daya, Qionghai, Xisha, and the SSCS) of the SSC. The dominant phylum is *Proteobacteria*, and the other two phyla, *Bacteroidetes* and *Firmicutes*, were relatively minor or rear components of the communities. Our results were consistent with earlier culture-independent studies that used 16S rRNA genes high-throughput sequencing in the SCS [37,38]. In fact, in the Veracruz Reef System (Southwestern Gulf of Mexico), it was also found that the dominant phyla of coral microbes were *Proteobacteria*, *Bacteroidetes*, *Firmicutes*, and *Cyanobacteria* [39]. In a study of bacteria diversity in the Great Barrier Reef, the coral symbiotic microorganisms *Alphaproteobacteria* and *Gammaproteobacteria*, including *Alteromona*, *Pseudoalteromonas*, *Halomonas*, and *Pseudomonas* were found to be dominant [40]. Coral-associated bacteria play significant roles in maintaining the health of coral reef ecosystems, material transformation, and biogeochemical cycles, which is also an important indicator of coral health [41,42]. *Proteobacteria* are a widely adapted group that exists in marine environments and represent the most important group of planktonic bacteria in coral reef ecosystems [43]. *Bacteroidetes* [44] are highly abundant in coral reef ecosystems, and have rich sources of saccharohydrolase, which can degrade a variety of marine polysaccharides [45]. Coral symbiotic bacteria are diverse and are associated with many environmental factors. 

*Alphaproteobacteria* and *Gammaproteobacteria* are considered the main classes of coral symbiotic bacteria [46]. Indeed, in this study, the number of identified *Gammaproteobacteria* (365 strains) absolutely dominated culturable heterotrophic bacteria from coral reef areas, followed by *Alphaproteobacteria* (51 strains). The coral mucus and surrounding seawater associated with the bacterial community in the SCS [2] and other coral reef areas [47] were found to be dominated by *Gammaproteobacteria*, which also carry out significant functions in nutrient cycling (e.g., carbon). For example, *Gammaproteobacteria* and *Alphaproteobacteria* [48] were found to be the most abundant nitrogen-fixing bacteria in some coral species, and the diseased corals usually contained a higher diversity of nitrogen-fixing bacteria than the healthy corals. On the contrary, in the coral reef systems of the Xisha Islands and Indo-Pacific [16,38], most *Proteobacteria* were affiliated with *Alphaproteobacteria*. Previous research has found that dominant species are replaced by *Gammaproteobacteria* as bleaching develops in the SCS [17]. Thus, we suspect that the higher proportion of *Gammaproteobacteria* compared to *Alphaproteobacteria* may be related to coral distribution. *Vibrio*, *Pseudoalteromonas*, *Alteromonas*, and *Flammeovirga*, which accounted for the majority of culturable bacteria, were the dominant genera. 

*Vibrio* spp. isolated from the four regions showed the highest diversity and abundance, especially the relative abundance of coral pathogens such as *V. owensii*, *V. campbellii*, and *V. harveyi*. Moreover, coral reef ecosystems were closely related to *Vibrio* abundance and diversity, strains such as *V. owensii* [25], *V. campbellii* [49], *V. harveyi* [50] are potential pathogenic strains that cause coral bleaching and other coral diseases, although not all strains of these species are pathogenic or have different levels of virulence [15]. In this study, *Vibrio* isolates from Qionghai showed the lowest abundance and diversity, with a small proportion of coral-pathogenic *Vibrios*. In contrast, a large number of coral-pathogenic *Vibrios* were isolated from the SSCS and Xisha Islands, such as *V. owensii*, *V. campbellii*, and *V. harveyi*. Previous studies have suggested that a *Vibrio*-dominated community commenced prior to visual signs of bleaching [51] and also reflected its strong competitive abilities on specific corals [52]. Thus, the difference in the abundance of pathogenic *Vibrio* may be related to coral bleaching in the SSCS and Xisha Islands, which were more severe than the other two sampling sites in 2020 [53]. Additionally, *Pseudoalteromonas* was the genus with the second largest number of strains, which could produce antibacterial compounds, toxins, bacteriolytic substances, and enzymes to take part in coral holobiont defense and be beneficial microorganisms for corals, helping to alleviate pathogen and temperature stresses [54]. Allers et al. suggested that *Alteromonas* and *Vibrio* played important roles in the organic carbon and nutrient transfer [55] from coral mucus to pelagic microbial food webs [56]. Thus, these dominant strains may greatly affect coral metabolism and material circulation, some of which may cause coral diseases and thus participate in the stability of coral reef ecosystems.

Meanwhile, the distributions of heterotrophic bacteria also showed distinct patterns among the four sampling areas. Various potential functional groups were essential to the coral reef, which may associate with the coral reef in terms of similar function, rather than identical species [27,57]. The area with the highest abundance and diversity of bacteria is the SSCS, where 158 strains belonging to 46 species and 17 genera were isolated. On the contrary, Daya Bay had the least abundance and diversity of bacteria, with 81 strains belonging to 20 species and 8 genera. The reason might be that the number and species of coral in the SSCS and DY may have significant differences, and it has been reported that bacteria usually show separated distributions in coral-rich (Xisha Islands) and coral-poor areas (Daya Bay) [58]. In addition, there are discrepancies in the related bacteria species among different coral species in the SCS, which may be related to the adaptation of coral to the marine environment with latitude variation. Corals have their own preferences for the selection of associated bacteria, and the abundance and diversity of bacteria can help corals respond to abnormal changes in temperature and play an important role in coral adaptation to large environmental fluctuations [2]. The primary productivity of coral reef areas is high, and many organic substances are released into the seawater in the form of coral mucus, allowing surrounding bacterial communities to rapidly respond to the enrichment of nutrients [59]. We speculate that the primary cause of the striking changes in bacterial communities in the SCS may be due to differences in physical processes, such as mesoscale eddies and river freshwaters, as well as changes in seawater chemistry, such as nutrient and salt concentrations [37]. However, *Actinobacteria*, as a ubiquitous major group in the coral holobiont [60], which were isolated in many coral reef area, were not found in this study. This may be due to Actinobacteria’s recalcitrant to standard culture methods, or they grow slowly on the media used in this study [61].

### 4.2. Coral Reef Areas May Be a Vast Treasure Trove of Novel Species and Agarases

The cycling of carbon [62] and nutrients and symbiotic relationships between corals and their associated microorganisms [63] alter microbial activity, which increases the abundance and diversity of microorganisms and microbiological degradation activities. Based on the special ecological environment of coral reefs, the number of potential novel species isolated in this study is relatively large. Nineteen potential novel species were isolated from coral reef areas in the SCS, accounting for 2.4% of the total. These potential novel species were of a wide variety, belonging to 10 genera, such as *Thalassotalea*, *Tencibaculum*, *Flammeovirga*, and *Flavivirga*, with the highest proportion of strains belonging to *Vibrio*. *Vibrio* is ubiquitous in marine and estuarine environments, with over 160 species validly described [64]. However, there are still a lot of potential novel species in the genus *Vibrio* spp. in coral reef waters. Furthermore, *Flammeovirga* is a relatively new genus, and only seven valid species have been included in the NCBI database [65]. Thus, coral reefs may provide special environmental conditions for the survival of microorganisms, which are richer in organic matters than non-reef areas [58].

Agar-degrading bacteria are crucial divers for the carbon cycle in the marine ecosystem [66]. Strains with agar-degrading activity can participate in the carbon cycle by using plankton and their debris as carbon sources, thus playing an important role in the regulation of coral reef ecosystems [67,68]. To understand the potential roles of heterotrophic bacteria in the coral reef ecosystem, a number of strains were tested for their agar-degrading abilities. Twenty-eight strains could degrade agar, belonging to *Vibrio*, *Pseudoalteromonas*, * Marinobacter*, *Alteromona*, *Flammeovirga*, and *Agarivorans.* Similarly, several studies have reported that most of the identified β-agarase (GH16) was produced by marine bacteria, such as *Pseudomonas* [69,70], *Pseudoalteromonas* [71,72], *Alteromona* [73], *Cytophaga* [74], *Vibrio* [75], *Agarivorans* [76], and *Flammeovirga* [77]. Significantly, all strains of *Flammeovirga*, such as *F. pacifica* [78], *F.* sp. OC4 [79], *F.* sp. SJP92 [80], and *F. yaeyamensis* AM5.A [81], showed endolytic activity on agar degradation. In this study, there were two agar-degrading genes in *F. agarivorans* SR4; *aga 1177* belonged to the GH16 protein family, and *aga 1178* was affiliated with the GH86 protein family. According to the homology of amino acid sequences, a lot of the known β-agarases are grouped into GH16, 50, 86, and 118 families [78], which are the main enzymes in the functioning of the extracellular agarase system, breaking the β-1,4 glycosidic bond to generate new agar oligosaccharides [82]. Meanwhile, strain SM6 was a member of the genus *Vibrio*, and several species within *Vibrio* have been found to have agar-degrading activity [83]. These findings can help expand the resource of strains with agar-degrading activity in *Vibrio* and better understand the degradation pathway of agar in marine ecosystems [84].

### 4.3. Whether TCBS Agar Is a Suitable Medium for the Isolation of Vibrios?

Vibrios tend to dominate the microbiota of environmental settings with high loads of nutrients and at high temperatures, especially some pathogenic *Vibrio* bacteria. Moreover, Rosenberg et al. have suggested bacteria could be the main cause of coral death, often caused by *Vibrio pathogens* [85,86]. Currently, research is underway to classify and identify the cultivable microbial communities associated with coral bleaching, including *Vibrio* and other heterotrophic bacteria, using cultivation methods and molecular biology techniques [27]. TCBS medium is well-known as a selection medium for *Vibrio*, and the vast majority of pathogenic vibrios grow well, such as *V. cholerae*, *V. parahaemolyticus*, *V. shiloi*, and *V. mimicus* [87,88]. In this study, 348 strains were isolated from TCBS medium, and 294 strains belonged to *Vibrio*, accounting for 84.5%. Thus, we considered that TCBS agar is a suitable medium for the isolation of vibrios, but the selectivity shows limitation. Indeed, other genera like *Photobacterium* (13 strains), *Pseudoalteromonas* (13 strains), *Halomonas* (11 strains), and *Shewanella* (4 strains) were isolated from TCBS agar in our study. Previous studies have reported that TCBS agar can support the growth of other genera and is not extremely selective for *Vibrio*. This highlights the extreme plasticity and heterogeneity of *Vibrio* members, thus confirming that assigning them precisely to a single species is unreliable [89,90]. The proportion of *Vibrio* bacteria varies with different sampling sites and sampling times [91]. Nevertheless, some studies have provided more accurate and likely estimations of the actual presence of vibrios by modified TCBS medium [92,93]. Thus, the isolation of *Vibrio* spp. still faces obstacles, and the combinations of culturable and molecular approaches should be applied in the future.

## 5. Conclusions

In summary, 441 heterotrophic bacterial strains (belonging to five classes, 43 genera, and 101 species) from the surrounding seawater near the coral reef areas (SCS) were isolated by MA medium. Due to the different sampling environments, the SSCS had the highest abundance and diversity of isolated bacteria, and Daya Bay had the rarest strains. *Gammaproteobacteria* was the dominant class, which might be associated with coral bleaching. Then, high coral pathogenic vibrios were found in the SSCS and Xisha Islands, which could be associated with coral bleaching. More potential novel species (19 species) and agar-degrading strains (28 strains) were recorded from the SCS than in earlier studies. Although culturable bacteria make up only a small proportion of all bacteria compared to culture-independent studies, isolated cultures can be used in studies such as the physiological properties of microorganisms. In the future, the overexpression and validation of agar-degrading strains would provide a better understanding of their adaptive mechanisms to marine environments, and the agarases may have potential value for industrial applications. 

## Figures and Tables

**Figure 1 microorganisms-12-00187-f001:**
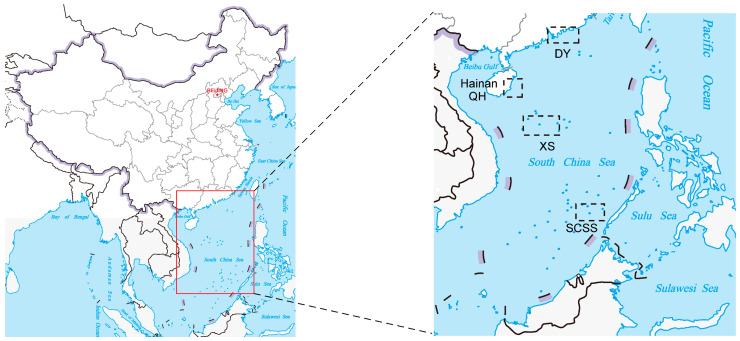
Map showing locations of the study area and sampling sites. (The map was modified from the website of standard map service, Ministry of Natural Resources, China [http://bzdt.ch.mnr.gov.cn; 16 August 2023]).

**Figure 2 microorganisms-12-00187-f002:**
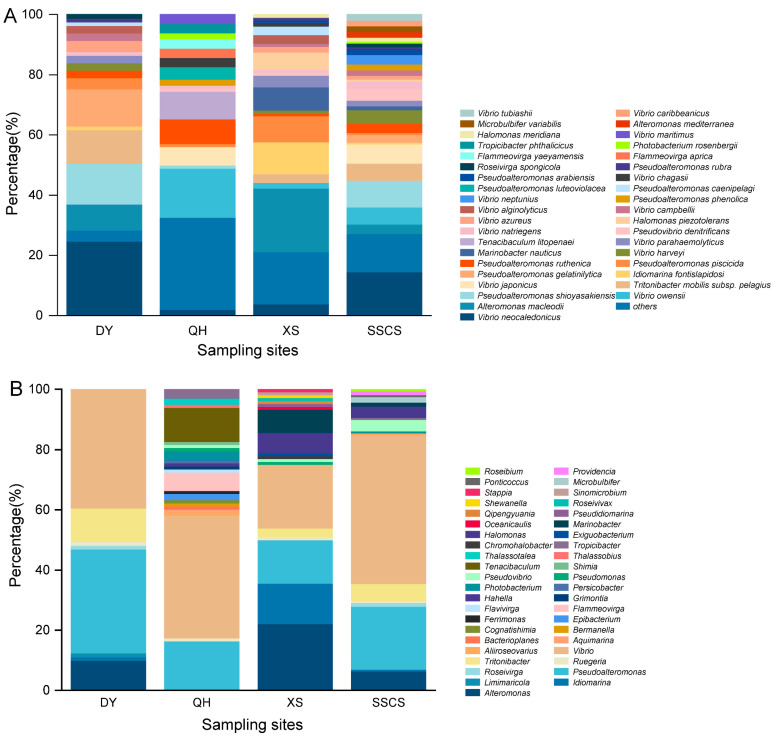
The diversities of total cultivable bacteria isolated from the SCS by marine agar. (**A**) At the species level; (**B**) at the genus level.

**Figure 3 microorganisms-12-00187-f003:**
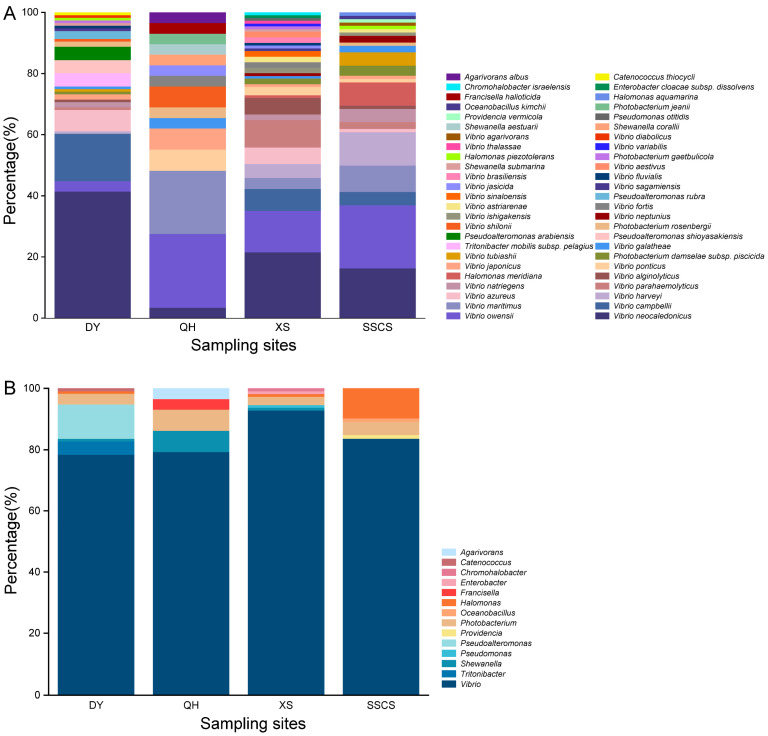
Bacterial diversity isolated from the SCS by TCBS agar. (**A**) At the species level; (**B**) at the genus level.

**Figure 4 microorganisms-12-00187-f004:**
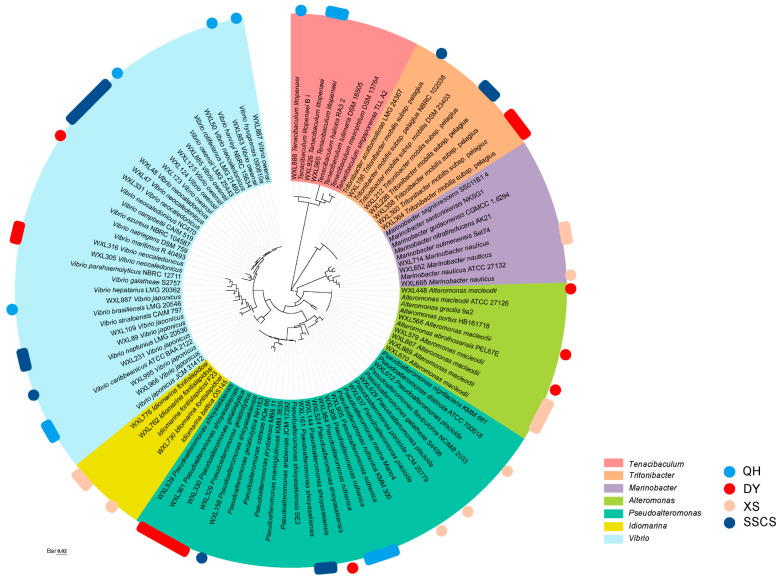
Neighbor-joining phylogenetic tree of the dominant bacterial members from coral regions. Bar, the number of substitutions per site.

**Figure 5 microorganisms-12-00187-f005:**
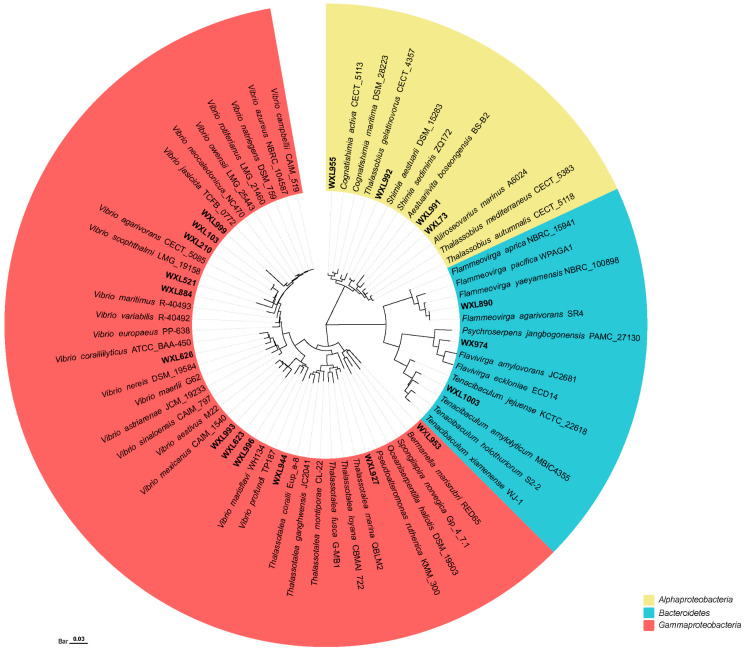
The neighbor-joining phylogenetic tree of potential novel bacterial strains among all samples. The top-hit taxon of potential novel strains is marked with bold font. Bar, the number of substitutions per site.

**Figure 6 microorganisms-12-00187-f006:**
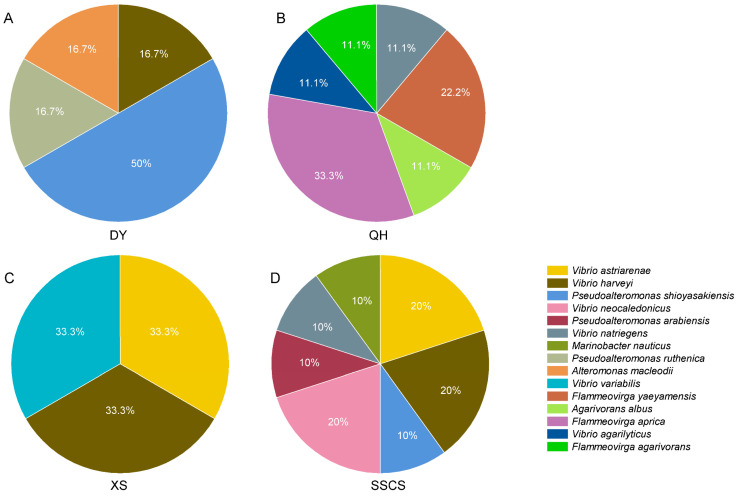
The relative abundances of agar-degrading bacteria. (**A**) Agar-degrading bacteria isolated from Daya Bay; (**B**) agar-degrading bacteria isolated from Qionghai; (**C**) agar-degrading bacteria isolated from the Xisha Islands; (**D**) agar-degrading bacteria isolated from the Southern South China Sea.

**Figure 7 microorganisms-12-00187-f007:**
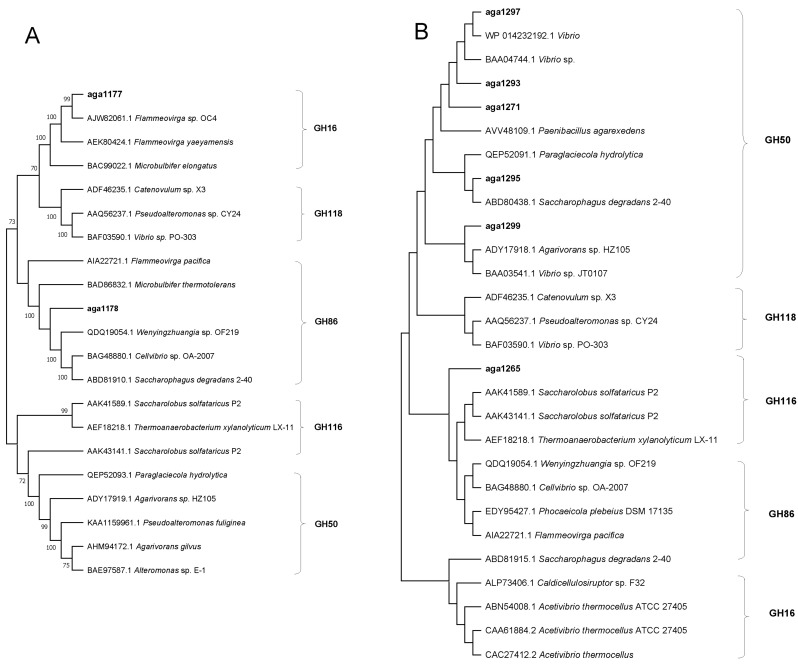
Phylogenetic analysis of agar lyase-encoding genes from strains WXL890 (SR4) and WXL884 (SM6) with other agarases based on the amino acid sequences. (**A**) Agar lyase-encoding genes from strain WXL890 (SR4); (**B**) agar lyase-encoding genes from WXL884 (SM6).

## Data Availability

The 16S rRNA gene sequences of the 789 isolates were listed in Appendix A. The Draft Genome Shotgun projects of strains SR4 and SM6 have been deposited at DDBJ/ENA/GenBank by our laboratory previously, under the accession numbers SAMN14599480 and SAMN14599479, respectively.

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
