# Peer review of "Diversity of Culturable Bacteria from the Coral Reef Areas in the South China Sea and Their Agar-Degrading Abilities"

_microorganisms, 2024, doi:10.3390/microorganisms12010187_

Round 1
Reviewer 1 Report
Comments and Suggestions for Authors
The authors isolated and analyzed the bacterial community structure of seawater samples and studied their characteristics and determined their role in the stability of the coral ecosystems.
In the field of determining the role of bacterial communities in the stability of the coral ecosystems.
Identified new species and studied agar degradability properties.
There is no additional methodology required
The conclusions consistent with the evidence and arguments
presented and do they address the main question posed.
1- Remove or substitute the keywords present in the title.
2- Line 95: Error! Reference source not found.?
3- southern south China sea (SCSS)?
4- In line 535: 10-1 to 10-3 with 10-1 to 10-3

Author Response
Response to the editor and reviewers (microorganisms-2822384)
Many thanks for the editor and reviewers’ comments. We are able to answer all the questions that the reviewers have proposed, and have carefully revised the manuscript in response to the reviewers (changes in the revised manuscript are highlighted in blue).
Here are point-by-point responses to the reviewers:
- Remove or substitute the keywords present in the title.
Reply:Thank you for the comment. We have changed the keywords, as follows: “Coral reef; Diversity; Heterotrophic bacteria; Vibrio; Agarases” (Line 37).
- Line 95: Error! Reference source not found.?
Reply: Thanks for your comments. We have checked and modified it (Line 64).
- southern south China sea (SCSS)?
Reply: Sorry for the confusing. We have checked and changed the “SCSS” to “SSCS” (Line 93).
- In line 535: 10-1 to 10-3 with 10-1 to 10-3
Reply: Thanks for your comments. We have changed the “10-1 to 10-3” to “10-1 to 10-3” (Line 108).

Reviewer 2 Report
Comments and Suggestions for Authors
Recommendations for authors on file

Author Response
Reviewer #2
In the presented paper "Diversity of culturable bacteria from coral reef areas in the South China Sea and their ability to degrade agar" by Mei Liu et al, the authors analysed the bacterial community structure of seawater samples from Daya Bay, Qionghai Bay, Xisha Islands and the southern part of the South China Sea based on culturing methods and revealed their ability to degrade agar.
The authors have done serious work to collect and analyse a large amount of material. The purpose of the study is relevant to the described region and is of interest to a wide audience of readers. Modern statistical methods have been used to analyse the data, the article is beautifully illustrated. However, the design of the work, its structure and writing of some sections do not allow to fully appreciate the full potential of this study.
Reply: Thanks for your comments. We have carefully revised the manuscript, and we have modified all the questions that you have proposed. And, the point-by-point responses have been listed as below.
- Unnecessary abbreviations have been introduced in the Abstract section and are not used hereafter in this section: In this study, we have analyzed the bacterial community structure of seawater samples among Daya Bay (Fujian Province, DY), Qionghai (Hainan Province, QH), Xisha Islands (XS)and southern South China sea (SSCS) based on the culturable methods and detected their abilities for agar degradation.
Reply: Thanks for your comments. We have removed the abbreviations in the Abstract, as follows: “In this study, we have analyzed the bacterial community structure of seawater samples among Daya Bay (Fujian Province), Qionghai (Hainan Province), Xisha Islands and southern South China sea based on the culturable methods and detected their abilities for agar degradation.” (Line 24-26).
- There is a lack of deciphering the abbreviation of TCBS agar.
Reply: Thanks for your comments. The “TCBS agar” means to “thiosulfate citrate bile salts sucrose agar”, and we have checked and modified it (Line 85).
- In addition, we would like to know their exact number: Strains belonged to the genus Vibrio had the greatest number, indicating the high selectivity of TCBS agar for vibrios.
Reply: Thanks for your comments. In our study, the number of Vibrio strains isolated from TCBS agar was 294 strains. And, we have added the exact number (Line 185).
- The Introduction section is written very stretched, I would recommend a more concise and clear statement of aims and objectives in the last paragraph, rather than some sort of results of the work done.
Reply: We appreciated the reviewer’s valuable comments. We have made changes to this section: The abundance and diversity of bacteria in coral reef areas of the south China sea (SCS) are higher than the non-coral areas, though also face coral disease and bleaching events currently [26]. The dominant bacteria of different coral species are similar, which were distributed in Gammaproteobacteria, Alphaproteobacteria, Firmicutes [27], Bacteroidetes and Actinobacteria. Coral bleaching may be attributed to the dominant Vibrio associated with corals [28]. Here, the seawater samples were collected near the coral reefs in the SCS, including Daya bay (DY), Qionghai (QH), Xisha islands (XS) and the southern south China sea (SSCS) during May to October 2020. We used marine agar 2216E agar (MA; Becton Dickinson) to isolate heterotrophic bacteria and TCBS agar to isolate coral pathogenic Vibrio. In addition, the bacteria were identified based on 16S rRNA gene sequencing, and further detected agar degradability. Our results will provide a unique microbial resource base for understanding the diversity of heterotrophic bacteria, and will help investigate their potential roles in the coral ecosystems (Line 88-97).
- It is unclear why the Materials and Methods section is not immediately after the introduction, but is presented at the end of the paper. I recommend following the classical structure of the manuscript. In addition, I was not provided with the S1 file and could not read it. From the Results section, transfer Figure 1 to the Materials and Methods section.
Reply: Thanks for your comments. We have taken the Materials and Methods section immediately after the Introduction (Line 98). And, we apologize for the missing of the table S1 file, and we have submitted the supplementary file (Table S1). In addition, we have transferred Figure 1 to the Materials and Methods section (Line 103).
- Conclusions: «Due to differences in sampling conditions, SSCS had the highest abundance and diversity of isolated bacteria, while Qionghai had the rarest strains». The text repeatedly mentions the differences in sampling but does not provide any supporting data: temperature, salinity, biogenes at the sampling points. Perhaps these data are available in appendix S1? Was PCA, RDA or ANOSIM analysed between DY, QH, XS, SSCS by environmental factors or by bacterial community? That would remove a whole host of questions.
Reply: Thanks for your comments. Due to the secrecy of environmental parameters in the SSCS, we are unable to provide detailed environmental parameters and do the correlation and dbRDA analyses. Based on previous research by our team, we think that the heterogeneity of bacteria among the four regions may reflect differences in environmental conditions between coral rich (SSCS) and coral poor environments (Daya Bay; Line 379-381). Meanwhile, we have added the reasons for our findings in the “Discussion” section, as follows: “The reason might be that the number and species of coral in the SSCS and DY may have significant differences, and it has been reported that bacteria usually showed separated distributions to coral-rich (Xisha Islands) and coral-poor areas (Daya Bay) [58]. In addition, there are discrepancies in the related bacteria species among different coral species in the SCS, which may be related to the adaptation of coral to the marine environment with latitude variation. Corals have their own preferences for the selection of associated bacteria, and the abundance and diversity of bacteria can help corals respond to abnormal changes of temperature and plays an important role in coral adaptation to large environmental fluctuations [2].” (Line 318-324). Additionally, we have done the PCA and ANOSIM analyses, and the results have showed that these four areas are discrete with significant differences (ANOSIM, P <0.01; showed in the figure below). PCA and ANOSIM analyses usually used for the results from high-throughput sequencing, our study focused on the comparison of culturable bacteria and the data among four sampling areas could not be normalization. Thus, we have not added the results in the main text. If you think we can put it in, we would add it in the manuscript. (We've put the figures in the file.)
Figure. Principal component analysis (PCA) based on compositional differences in bacteria from four regions. A, four areas: DY, QH, XS, NS; B, different stations in the four regions.

Round 2
Reviewer 2 Report
Comments and Suggestions for Authors
The authors have done a lot of work to improve the manuscript and it can be accepted in its present form